



# Active Trailing Edge Flap System fault detection via Machine Learning

Andrea Gamberini[1,2] and Imad Abdallah[3]

[1]Siemens Gamesa Renewable Energy A/S, Brande, Denmark
[2]DTU, Dept. of Wind and Energy Systems, Roskilde, Denmark
[3]Chair of Structural Mechanics and Monitoring, ETH Zurich

**Correspondence:** Andrea Gamberini (andgam@dtu.dk)

**Abstract.** Active trailing edge flap systems (AFlap) have shown promising results in reducing wind turbine (WT) loads. Once the WT design includes the AFlap, a condition monitoring system will be needed to ensure the flaps provide the expected load reductions. This paper presents two approaches based on machine learning to diagnose the health state of an AFlap system. Both approaches rely only on the sensors commonly available on commercial WTs, avoiding the need and the cost of additional measurement systems. The first approach (MFS) uses manual feature engineering in combination with a random forest classifier. The second approach (AFS) relies on random convolutional kernels to create the feature vectors. The study shows that the MFS method is reliable in classifying all the investigated combinations of AFlap health states in the case of asymmetrical flap faults not only when the WT operates in normal power production but also before startup. Instead, the AFS method can identify some of the AFlap health states for both asymmetrical and symmetrical faults when the WT is in normal power production.

## 1 Introduction

The pursuit of lower Levelized Cost of Energy has driven a steady increase in the size of utility-scale Wind Turbines (WTs) over the past years, with a consequent increase in the load carried by the WT components. Among the new technologies studied to mitigate this load increase, actively controlled flaps located at the blade trailing edge (AFlap) have shown promising results in reducing fatigue and ultimate loads and increasing annual energy production, see Barlas et al. (2016), and Pettas et al. (2016). Despite the potential benefits of AFlaps, this technology has yet to reach a sufficient level of maturity for its implementation in commercial WTs. To the authors' knowledge, only Siemens Gamesa Renewable Energy (SGRE) has publicly shared data of an AFlap system implemented on two different multi-MW WTs: a 4.0 MW WT prototype and a 4.3 MW WT prototype, both installed in Høvsøre (Denmark), see Gomez Gonzalez et al. (2022).

Once the AFlaps reach a sufficient maturity level to be integrated into the WTs' design, a fault diagnosis and condition monitoring of the AFlap system will be needed to ensure the AFlaps system provides the expected load reductions. This monitoring ability will be critical to ensure the WT's performance and integrity. Until now, the fault diagnosis and condition monitoring of AFlap systems have not been detailed investigated, and to our knowledge, no literature is available on this topic. Nevertheless, we can foresee different approaches for AFlap fault diagnosis and monitoring, following the standard methodologies currently





applied in the wind energy sector.

First, monitoring and diagnosis can rely on dedicated sensors located in specific mechanical elements, like a temperature sensor in a gearbox. For the AFlap system, position or pressure sensor could be located on the flap surfaces or in their proximity to quantify the AFlap deflection or the AFlap impact on the blade aerodynamic. Due to the expected large blade area covered by the flaps, this monitoring approach will require several sensors distributed along the outer third of the blade length. This

system will likely be complex, expensive to deploy and maintain, sensitive to lighting, and affected by the reliability of the sensor operating in the harsh environment of a wind turbine rotating blade.

The second approach is the model-based method that mainly relies on the analyses of the residual signals, signals defined as the difference between the real system outputs and the output from a model of the system created by using, for example, Kalman filter, observers, or model based machine learning techniques. On WTs, the model-based methods have been applied,

for example, in the condition monitoring of main bearing (de Azevedo et al., 2016), sensor and actuators (Cho et al., 2018), and generator (Gálvez-Carrillo and Kinnaert, 2011). As a drawback, the model-based methods require a reasonably good model to guarantee the detection of faults. The model generation could be challenging for AFlap fault detection, mainly due to the high nonlinearity of the WT blade dynamic, the high uncertainty on the wind field interacting with the WT rotor, and the limited number of sensor measurements available on a commercial WT. Improved wind field estimations (e.g., from nacelle Lidar) or

additional load or pressure sensors on the blade can facilitate the model generations and improve their accuracy at the price of an increased system complexity and cost.

Finally, data-driven methods allow fault detection without needing a detailed system model but by different types of data and signal analysis. These analyses range from the simple detection of changes in mean values, variances, or trends to the more advanced machine learning (ML) methodologies (Badihi et al., 2022). In particular, the study and application of ML method-

ologies to fault diagnosis and condition monitoring has increased exponentially in the recent years thanks to the technological and computational advances that have allowed to quickly and efficiently analyze the large amount of data needed for the training of the ML models (García Márquez and Peinado Gonzalo, 2022). An overview of the Machine learning methods for wind turbine condition monitoring is provided by (Stetco et al., 2019). ML techniques can be applied for the AFlap fault detection if a sufficient amount of relevant data can be provided for the model training. Currently, the amount of AFlap field data is limited,

even more for AFlap faults. Nevertheless, Aeroelastic simulations have been commonly used for WT design, therefore it is reasonable to assume a sufficiently accurate aeroelastic model of a WT equipped with AFlap can be used to train a ML model for the AFlap fault detection. To test this assumption, in this paper we study if a data-driven methods based on ML trained with aeroelastic simulation can properly classify the AFlap fault states.

### 1.1   Detecting AFlap health state

The detection of the health state of the AFlap is a challenging task. Figure 1a shows the mean blade root moment when the flap is deactivated (AF_Off) or is active without performance degradation (AF_On) in function of the wind speed. As expected, AFlaps have a relevant impact on the WT blade aerodynamic, visible in the two distinct lines of the mean moment binned in function of the wind speed. However, the broad range of environmental conditions where the WT operates causes the moment

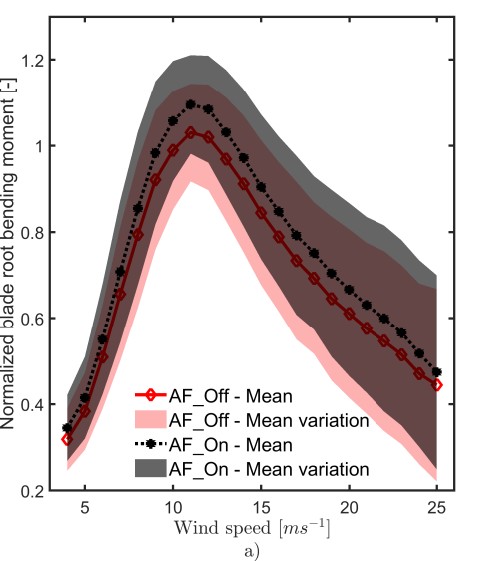 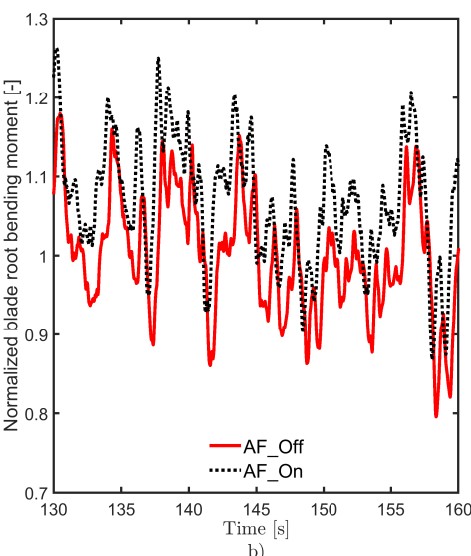

**Figure 1.** a) Mean value of the normalized blade root bending moment when the AFlap is deactivated (AF_Off) and activated without degradation (AF_On), binned in function of the wind speed. b) Example of time series of the normalized blade root bending moment when the AFlap is deactivated (AF_Off) and activated without degradation (AF_On) for a $10\ ms^{-1}$ turbulent wind

to vary within a wide range of values, range that is shown in the plot by the colored areas. These areas overlap significantly, making it difficult for a detection system to identify the actual AFlap health state. Figure 1b shows the time series of the blade root bending moment for the AFlap active and AFlap not active with the same $10\ ms^{-1}$ turbulent wind. The two lines have a similar highly oscillating behavior with just a small shift due to the increased lift generated by the flap activation.

Furthermore, several faults of the AFlap system can occur in the WT lifetime, which behavior and severity depend on the layout and scope of the AFlap system itself. Wear and tear can slowly degrade the performance of part of the AFlap system; ice or lightning can instead compromise the whole system's functionality. As it is impossible to test all the different combinations of faults, we selected a set of representative conditions we believe can cover a wide range of flap faults. The selected cases cover partial and complete performance degradation happening only on one blade or on all three blades simultaneously. To keep the approach as general as possible, we focus on identifying the AFlap health state in static flap actuation. This approach keeps the detection system independent from any specific AFlap controller strategy, AFlap system design, or Fault dynamics. The idea is to integrate this kind of detection system in an AFlap status check routine running for several minutes where the performance of the stationary flap is verified.

## 1.2 Research contribution

This paper investigates whether a simple ML algorithm can assess the health of an active trailing edge flap system from the data provided by the sensors commonly available on a commercial WT. The aim is to develop a system that does not require



any additional sensor to be installed on the WTs, making it easy to implement without relevant additional costs for installation or maintenance. This task can be seen as a multivariate time-series classification problem where the ML algorithm aims to estimate if the AFlap is properly operating or is affected by performance degradation. We follow two different approaches for computing the features from the sensors' time series data. In the first approach, we manually select the features based on our knowledge of the impact of AFlaps on the different WTs signals. In the second approach, multiple random convolutional

kernels automatically generate the features from all the available signals, without requiring pre-knowledge of the AFlaps' impact on the WT signals. We select the simple but robust random forest classifier for both feature calculation approaches. Aeroelastic simulations are used to train and test the ML models. We use the aeroelastic model developed by (Gamberini et al., 2022) of the 4.3MW WT prototype owned by SGRE, where a 20m AFlap was installed and tested on one blade of the 120 m diameter rotor.

Section 2 describes the aeroelastic model, the environmental conditions, and the flap health states used in the aeroelastic simulations. In Section 3, we describe the ML methodologies used in the study and the two approaches used for AFlap health detection. In Sections 4 and 5, we show the obtained results that we discuss in Section 6.

## 2  Simulated experiments

The training of the ML models is based on a pool of aeroelastic simulations reproducing the WT aeroelastic response for the

combination of wind turbine operative conditions and flap health states of interest.

### 2.1  Aeroelastic simulations

A set of aeroelastic simulations is computed for every combination of wind turbine operative conditions and flap health state of interest. We computed all sets with the same WT aeroelastic model.

We accounted for the influence of the variability of the environmental conditions on the wind turbine's aeroelastic response by

defining the main environmental conditions as random variables of pre-imposed statistical properties.

#### 2.1.1  Environmental conditions

Table 1 shows the environmental conditions modeled as random variables and their parameters, which are:

- **Mean wind speed**: follows a Weibull distribution with the annual average wind speed set to 10 $ms^{-1}$ and the shape parameter to 2. It is equivalent to IEC wind class 1.

- **Wind turbulence intensity**: follows the normal turbulence model described in the IEC 61400-1:2019 (IEC, 2019) for turbulence class A where Iref is set to 0.16 with a lognormal distribution. It is defined as:

$$E[\sigma_U|U] = I_{ref}(0.75U + 3.8) \tag{1}$$

$$Var[\sigma_U|U] = (1.4I_{ref})^2 \tag{2}$$



**Table 1.** Parameters of the environmental conditions modelled as random variables

| Value | Unit | Distribution | Mean | Variance | Min | Max |
|-------|------|--------------|------|----------|-----|-----|
| Turbulence intensity | % | lognormal | Eq.(1) | Eq.(2) | - | - |
| Wind shear exponent | - | normal | Eq.(3) | Eq.(4) | -0.2 | 0.4 |
| Air density | $Kg/m^3$ | normal | 1.225 | 0.05 | 1.103 | 1.348 |
| Horizontal inflow angle | deg | normal | Eq.(5) | Eq.(6) | -6 | 6 |
| Vertical inflow angle | deg | normal | 3 | 3 | -2 | 8 |

– **Wind shear exponent**: the vertical wind profile is modeled with a power law with exponent $\alpha$. As proposed by Dimitrov et al. (2015), the wind shear exponent is normally distributed and conditionally dependent on the mean wind speed $U$ as follows:

$$E[\alpha|U] = 0.088(ln(U) - 1) \tag{3}$$

$$Var[\alpha|U] = \left(\frac{1}{U}\right)^2 \tag{4}$$

$\alpha$ values are constrained within realistic limits.

– **Horizontal inflow angle**: $\Psi$ follows a normal distribution, is truncated within realistic limits, and is conditionally dependent on the mean wind speed U as proposed by Duthé et al. (2021):

$$E[\Psi|U] = ln(U) - 3 \tag{5}$$

$$Var[\Psi|U] = \left(\frac{15}{U}\right)^2 \tag{6}$$

– **Vertical inflow angle** and **Air density**: normally distributed, truncated within reasonable limits, with prescribed values of mean and standard deviation.

One example set of environmental condition is sampled for the pre-startup cases, showed in Figure 2, and the normal power production cases, showed in Figure 3.

### 2.1.2 Aeroelastic model and simulation setup

BHawC is the aeroelastic engineering tool developed internally by SGRE. It is based on the Blade Element Momentum (Fisker Skjoldan, 2011) and models the AFlap's aerodynamic and actuator system with a dedicated flap module.

SGRE provided the BHawC model of the prototype wind turbine (pWT) used for the aeroelastic simulations. It includes the pWT's structural and aerodynamic models and its controller. The AFlap model is also tuned to match the lift increase in the blade region covered by the flap when activated. In Gamberini et al. (2022), one of the authors showed that this model estimates the pWT operational parameters and blade loads with reasonable accuracy, and the AFlap activation increases up to +8% the

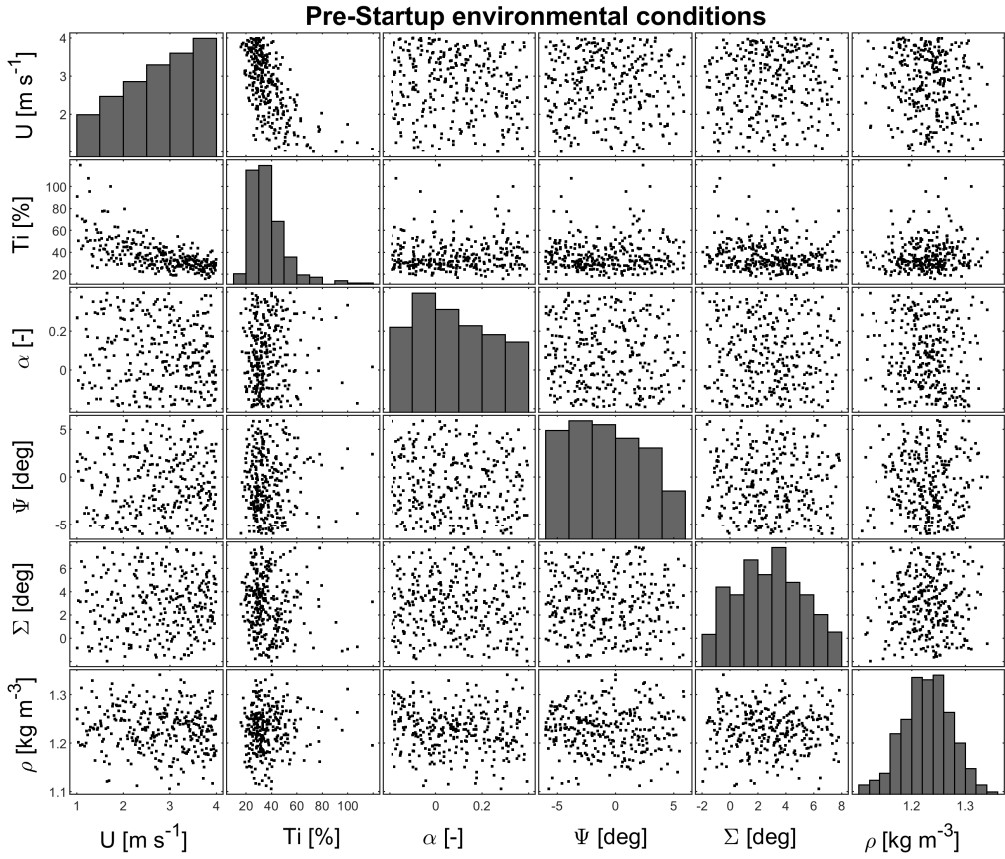

**Figure 2.** Example of environmental conditions for Pre-Starup simulations: mean wind speed U, wind turbulence intensity TI, wind shear exponent $\alpha$, horizontal inflow angle $\Psi$, vertical inflow angle $\Sigma$, and air density $\rho$.

mean flapwise bending moment at the blade root. All simulations have the same pWT BHawC model, with only changes in the environmental conditions and the AFlap health state. The simulations are performed with turbulence wind and are 10 minutes long with 0.01 s time step length.

Two operative conditions are simulated: normal power production (NPP) and pre-startup (PS). In the latter cases, the wind turbine is in idling condition with the controller optimizing the blade pitch angle to bring the rotor speed and generator torque

to the startup conditions.

For every operative condition, asymmetrical and symmetrical AFlap fault cases are simulated. In the symmetrical flap fault case (3B), the flaps on the three blades have all the same state and performance. This means the three flaps are modeled with the same aerodynamic polars and control signal. In the asymmetrical case (1B), the AFlap is active only on one blade. Even if this is not an expected configuration for future wind turbines, this setup mirrors the pWT setup and in the future it can be

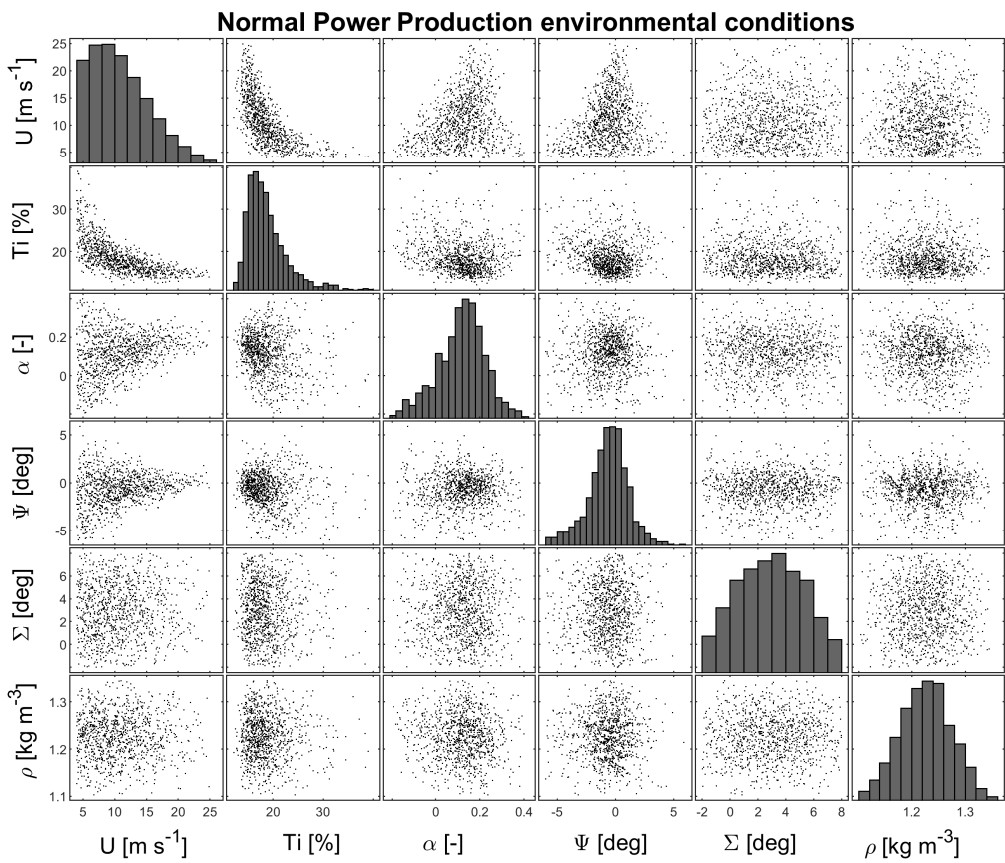

**Figure 3.** Example of environmental conditions for Normal Power Production simulations: mean wind speed U, wind turbulence intensity TI, wind shear exponent $\alpha$, horizontal inflow angle $\Psi$, vertical inflow angle $\Sigma$, and air density $\rho$.

tested with the pWT measurements data. Furthermore, the rotor imbalance due to the flap activated on only one blade is still a good approximation of the imbalance due to a flap on one blade being at a different state or performance than the flaps on the other two blades.

For every case, we simulated seven different AFlap health states:

– **Flap Off (AF_Off)**: AFlap not active, simulated with baseline aerodynamic polar and flap control Off.

– **Flap On (AF_On)**: AFlap active, simulated with active flap aerodynamic polar and flap control On.

– **Flap Off with fault (AF_Off_Fault)**: AFlap active even if the control commands it to be not active. This state can simulate the case when ice formed on the blades prevents the flap from closing. It is simulated with active flap aerodynamic polar and flap control Off.





    – **Flap On with fault (AF_On_Fault)**: AFlap not active even if the control commands it to be active. This state can be caused by ice preventing the flap from opening or the flap actuator not working. It is simulated with baseline aerodynamic polar and flap control On.

    – **Flap On with degradation**: AFlap active but with degraded performance. Reduced flap deflections due to reduced flap actuator operation, material aging, or extremely low temperature can be associated with these cases. We simulated AFlap performance reduced to 25% (AF_On_25pc), 50% (AF_On_50pc), and 75% (AF_On_75pc) by using a corresponded aerodynamic polar linearly interpolated between the baseline polar and the active flap one meanwhile the flap control is On.

In the simulations, the AFlap's health state is kept constant as this study aims to identify the stationary AFlap health state and not the exact time of the change of state a fault can trigger. Figure 4 shows an example of the normalized lift coefficient of the flap baseline (AF_Off, line with triangle), flap active (AF_On, line with squares),and flap active with performance reduced to 25% (AF_On_25pc, dashed line), 50% (AF_On_50pc, circles), and 75% (AF_On_75pc, dotted line).

For every AFlap health states computed in the NPP case, we make two simulation sets: a Training and Test (TaT) set of 1000 simulations and a Validation (Val) set of 500 simulations. The TaT sets cover a wind speed range from 3.5 and 25 $ms^{-1}$ and share the same sample set of environmental conditions. The Val sets cover the same wind speeds as the TaT sets but share a set of environmental conditions uncorrelated to the TaT set. The PS cases have a similar setup, but they covers wind speeds between 1 and 3.5 $ms^{-1}$; TaT sets have 300 simulations, and the Val sets 150. Environmental conditions sets used for the PS simulations are also uncorrelated to the NPP ones.

As the input of the ML model, we selected only signals commonly available in the modern commercial wind turbines. These signals are the pitch angle [deg], the rotor speed [rpm], the generator power [kW], flapwise and edgewise bending moments at the root of each of the three blades [kNm], and linear tower top accelerations [$ms^{-2}$] together with the flap actuator control signal [logic].

## 3 AFlap health state estimation with ML

### 3.1 Introduction

This paper investigates whether a simple ML algorithm can estimate the health state of an active trailing edge flap from the data provided by the sensors commonly available on a commercial WT. We approached this task as a multivariate time-series classification problem where the ML algorithm aims to estimate the AFlap health state. We followed two different approaches for computing the features from the sensors' time series data. The first approach relies on the manual selection of the channels and their relevant statistics that, from the authors' knowledge, are known to be impacted by the trailing edge flap system. In the second approach, multiple random convolutional kernels automatically generate the features from all the available signals. Based on the MiniRocket algorithm, this approach does not require pre-knowledge of the AFlap system's impact on the WT



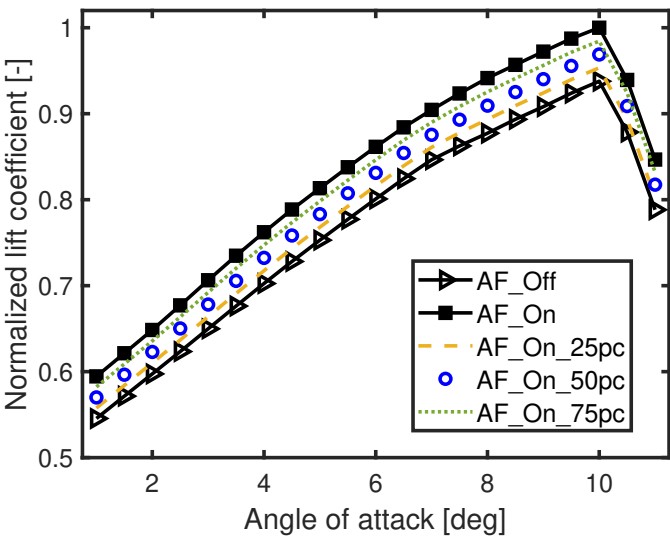

**Figure 4.** Example of flap normalized lift coefficient of the baseline (AF_Off, line with triangle), flap active (AF_On, line with squares),and flap active with performance reduced to 25% (AF_On_25pc, dashed line), 50% (AF_On_50pc, circles), and 75% (AF_On_75pc, dotted line).

signals and explores possible unknown relations among the different WT channels. As a classifier method, we selected the simple but robust random forest classifier for both feature calculation approaches. For the approach based on the MiniRocket algorithm, we also used the Ridge classifier with Cross Validation.

### 3.1.1 Random Forest

180 A Random Forest Classifier (RF) is a supervised discriminative machine learning technique whose objective is to estimate $P\left(Y \mid X, \theta\right)$ in which Y: target, X: observable, and $\theta$ are the parameters. We assume a multi-class classification problem where each observational sample is assigned to one and only one label, as opposed to the multi-label approach.

The Random Forest classifier is based on a collection of Decision Trees (DT, also called Classification or Prediction Trees), a non-parametric supervised learning method designed for the classification or regression of a discrete category from the data.

185 In the machine learning sense, the goal is to create a classification model (classification tree) that predicts the value of a target variable (also known as label or class) by learning simple decision rules inferred from the data features (also known as attributes or predictors). From Figure 5a, an internal node $N$ denotes a test on an attribute, an edge $B$ represents an outcome of the test, and the Leaf nodes $L$ represent class labels or class distribution. A decision tree is a tree-structured classifier built by starting with a single node that encompasses the entire data and recursively splitting the data within a node, generally into two

190 branches (some algorithms can perform $multiway$ splits). The splitting is obtained by selecting the variable (dimension) that best classifies the samples according to a split criterion, i.e., the one that maximizes the information gain among the random sub-sample of dimensions obtained at every point. The splitting continues until a terminal leaf is created by meeting a stopping





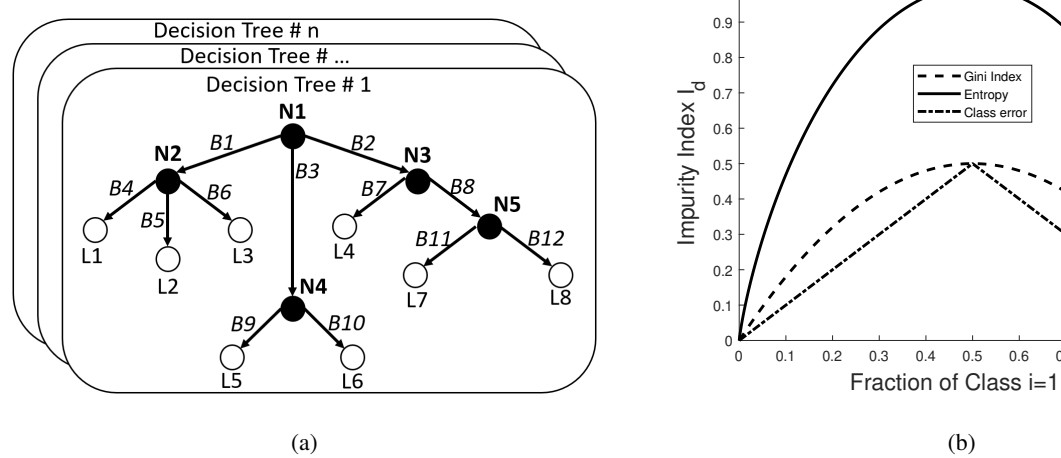

(a)

(b)

**Figure 5.** (a) Graphical representation of a forest of decision tree classifiers. (b) Impurity index $I_d$ for a two-class example as a function of the probability of one of the classes $f_1$ using the information entropy, Gini impurity and classification error. In all cases, the impurity is at its maximum when the fraction of data within a node with class 1 is 0.5, and zero when all data are in the same category.

criterion, such as a minimum leaf size or a variance threshold. Each terminal leaf contains data that belongs to one or more classes. Within this leaf, a model is applied that provides a reasonably comprehensible prediction, especially in situations where many variables may exist that interact in a nonlinear manner, as is often the case on wind turbines (Carrasco Kind and Brunner, 2013). Several algorithms exist for training decision trees with variations on impurity, pruning, stopping criteria, how to treat missing variables, etc. A top-to-bottom construction of a decision tree begins with a set of objects. Each object has an assigned label and a set of measured features. The dataset is split at each tree node into two subsets, left and right, so the two resulting subsets are more homogeneous than the set in their parent node. To do so, one must define a cost function that the algorithm will minimize. One can use the information entropy, the Gini impurity, or a classification error for the cost function. Formally, the splitting is done by choosing the attribute that maximizes the Information Gain, defined in terms of the impurity degree index $I_d$ as shown in Figure 5b.

By definition, a random forest classifier is a non-parametric classification algorithm consisting of a collection of decision tree-structured classifiers $\{h(\mathbf{x}, \Theta_k), k = 1, ...\}$ where the $\Theta_k$ are independent identically distributed random vectors, and each tree casts a unit vote for the most popular class at input $\mathbf{x}$. The RF prediction consists of the aggregation of the DT results obtained by a majority vote. Furthermore, the fraction of the trees that vote for the predicted class serves as a measure of certainty of the resulting prediction. RF improves prediction accuracy over single decision tree classifiers by injecting randomness that minimizes the correlation $\overline{\rho}$ amongst the grown individual decision trees $h(\mathbf{x}, \Theta_k)$ that operate as an ensemble. This is achieved by using bootstrap aggregating (a.k.a. bagging, sample with replacement of the training dataset) in tandem with random feature selection in the process of growing each decision tree in the ensemble Breiman (1996). This forces even more





variation amongst the trees in the model (different conditions in their nodes and different overall structures) and ultimately results in lower correlation across trees and more diversification.

Some of the arguments in favour of using a RF in this research include:

- Random Forests work well with both categorical and numerical data. No scaling or transformation of variables is usually
necessary.

- Random Forests implicitly perform feature selection and generate uncorrelated decision trees. It does this by choosing a random set of features to build each decision tree. This also makes it a great model when you have to work with a high number of features in the data.

- Random Forests are not influenced by outliers to a fair degree. It does this by binning the variables.

- Random Forests can handle linear and non-linear relationships well.

- Random Forests generally provide high accuracy and balance the bias-variance trade-off well. Since the model's principle is to average the results across the multiple decision trees it builds, it averages the variance as well.

- Random Forests are fairly interpretable. They provide both feature importance and in certain instances the ability to trace branches to follow the decision making process.

**3.1.2 MiniRocket**

ROCKET (RandOm Convolutional KErnel Transform) is an algorithm that generates a large number of convolution kernels (10000 by default) with random length, weights, bias, dilation, and padding of the time series provided as input. ROCKET extracts two features for each kernel: the maximum value (an equivalent to the maximum global pooling) and the proportion of positive values (PPV), indicating the proportion of the input matching a given pattern. The PPV is the most critical element

of ROCKET for achieving the state of the art accuracy (Dempster et al., 2020).

MiniRocket (MINImally RandOm Convolutional KErnel Transform) is a reformulated version of ROCKET (Dempster et al., 2021), 75 times faster while maintaining the same accuracy. MiniRocket minimizes the number of options for hyperparameters and computes only the PPV, generating half number of features as ROCKET. In addition, it does not require normalization. The v0.3.4 version of the MiniRocket implementation in (Oguiza, 2022) has been used in this paper.

**3.1.3 Ridge classifier with Cross Validation**

This Ridge classifier uses the Ridge regression to predict the class of a multiclass problem by solving the problem as a multi-output regression where the predicted class corresponds to the output with the highest value. Ridge Regression, also known as Tikhonov regularization, solves a regression model by minimizing the following objective function:

$$\min_{w} ||Xw - y||_2^2 + \alpha ||w||_2^2 \tag{7}$$



Where: $X$ are the training data, $y$ the target values, $k$ the ridge coefficients to be minimized, and $\alpha$ the regularization strength. $\alpha$ controls the amount of shrinkage: the higher the value, the greater the amount of shrinkage, increasing the robustness of the ridge coefficients to collinearity. The addition of the Cross Validation helps to identify the best set of ridge coefficients, reducing the risk of overfitting.

### 3.2   Manual Feature selection with Random forest classifier

The main effect of the activation or deactivation of a trailing edge flap on a WT is the change of local blade lift that consequently affects the blade's aerodynamic loading. The impact on the blade loading depends significantly on the WT operative conditions, as shown in Gomez Gonzalez et al. (2022). Furthermore, asymmetric activation of the flaps on the three blades leads to a rotor bending moments imbalance that is often associated with tower top vibration. Based on this, we manually select a set of signals to generate the features to train an RF model. In addition, we consider only signals commonly available in all modern
commercial wind turbines. The aim is the development of a method that can be implemented on commercial wind turbines without the need of installing additional hardware. Initially, we use simple statistical properties of the signal time series as features. Afterward, we include the Catch22 collection (Lubba et al., 2019) to expand the features pool.

### 3.2.1   Signal and features selection

The selected WT signals are:

– **Flap actuator control signal** [logic]: control signal of the flap activation state (On or Off).

    – **Flapwise and edgewise bending moments at the three blade roots** [kNm]: main signals to detect the impact of the flaps on the blade aerodynamic loading. Load sensors placed at the blade root are commonly available in modern WTs.

    – **Pitch angle** [deg], **rotor speed** [rpm] and, **generator power** [kW]: main signals to estimate the WT's operative condition to which the flaps' load impact is related. These signals are available in every WT controller.

– **Fore-Aft and side-side tower top accelerations** [$ms^{-2}$]: useful signals in detecting possible rotor imbalances. The WT safety systems commonly monitor these signals to detect WT anomalies.

    – **Out of plane rotor bending moments** [kNm]: signals computed from the blade root moments, pitch angle, and azimuth position through the Coleman transformation Bir (2008) equation. These moments help detect possible rotor imbalances.

The initially selected features are the standard deviation, mean, maximum, minimum, range, and maximum absolute value of
every signal. Afterward, we add the Catch22 collection (Lubba et al., 2019) to expand the features pool to explore possible unknown correlations between the input signals. We choose the Catch22 (CAnonical Time-series CHaracteristics) collection as it is a high-performing subset of 22 features selected over a pool of over 7000 based on their classification performance across a collection of 93 real-world time-series classification problems. The v0.4 Catch22 Matlab tool is used for this paper. Finally, we include the blade-to-blade difference of the mean and absolute maximum blade root bending moments to help detect possible





flap activation imbalances.

The features generation process computes circa 400 features for each aeroelastic simulation. To reduce complexity, we add two features filtering processes to the algorithm:

- **manual selection** of the desired feature subset in the algorithm pre-processing.

- **automatic features reduction** based on Out-of-bag permuted predictor importance (oobPPI) value. The oobPPI mea-
sures how influential a feature is in the model prediction by permuting the value of the feature and measuring the model error. The permutation of an influential feature should have a relevant effect on the model error; little to no effect should come from a permutation of a not influential feature. If this filter is active, the oobPPI of each feature is evaluated after the RF model is trained. Features with an oobPPI value below the threshold specified by the user are removed, and the RF model is trained again with the remaining features subset. The process is repeated until all the remaining features con-
sistently have an oobPPI value above the threshold, simplifying the final model by removing the features not influential on the classification process.

We decided to do not include the wind speed sensor signal in the ML training. This sensor is generally located on the nacelle, behind the rotor, where the complexity of the 3D wind flow can only be correctly estimated by high-fidelity codes, like CFD. In this paper we use a mid-fidelity aeroelastic model that is unable to estimate the wind speed on the nacelle with sufficient
accuracy. Therefore, training the ML model with a low-fidelity nacelle wind speed would reduce the model accuracy and performance on a real WT. Furthermore, the ML model can still derive the wind speed data from the rotor speed, pitch angle, and generator power, which strongly correlate to the wind speed. Therefore, omitting the wind speed signal reduces model uncertainties without losing relevant data for the ML training. Instead, we use the mean wind speed for splitting the NPP into different wind speed regions for which a dedicated RF model is trained. Generally, a modern WT operates differently below
rated wind speed, where it is power or torque controlled, compared to above rated wind speed, where it is pitch controlled. Therefore, we expect RF models trained for each specific wind region to perform better than a single RF model covering the whole wind speed range.

### 3.2.2   Algorithm structure and setup

The algorithm has the following structure:

1. Calculation of the features for every simulation. Usually it is done only once, after the aeroelastic simulation is computed.

2. Selection of the AFlap fault symmetry (1B or 3B), WT operative condition (NPP or PS).

3. Selection of the AFlap health states to be used in the classification.

4. For every specified wind speed range:

    (a) Manual selection of the desired feature subset.





(b)  Training of the RF model with the TaT set.

(c)  If the "automatic features reduction" is enabled, the following steps are repeat until all features have oobPPI above
threshold:

i.  Evaluation of the oobPPI of every features.

ii.  Removal of the features with oobPPI below threshold.

iii.  Training of a new RF model with the remaining features.

(d)  Validation of the trained RF model with the Val sets.

In the algorithms, three Matlab functions are used: **templateTree** to create the decision tree template; **fitcensemble** to train the
Rf model, and **oobPermutedPredictorImportance** to compute the oobPPI value for each feature.

We use the setup proposed by (Abdallah, 2019) as default RF setup: number of trees of 100, learning rate of 0.25, maximum
number of split of 12, test rate of 30% and impOOB threshold of 0.01.

### 3.3   Automatic Feature Generation with Random Forest or Ridge classifier

The AFS approach relies on the same signals used for the manual feature selection approach: pitch position, rotor speed,
generator power, flapwise and edgewise bending moments at the root of each of the three blades, linear tower top accelerations,
and the flap actuator control signal. As described in Section 3.2, these signals are relevant to detect the flap impact on the WT
and are provided with the standard sensors available on commercial WT. Instead of generating a set of features for each signal
based on statistical properties, the AFS approach utilizes ML techniques developed for image processing to create features of
the whole simulation. We implement two different algorithms for the classification: a RF classifier, similarly on what we used
for the MFS, and a Ridge classifier with Cross Validation (Ridge) suggested by (Dempster et al., 2020) for application with
MiniRocket.

#### 3.3.1   Feature generation with MiniRocket

MiniRocket works by first combining the time series of the relevant signals of a single simulation in a single matrix, aligning
them in function of time. Then it processes the resulting matrix like an image utilizing a kernel from which the proportion of
positive values is computed. A set of 10000 kernels of random length, weights, bias, dilation, and padding, are used, generating
10000 features per simulation. This process is repeated for all the simulations. For consistency, the same kernel set must be
used for all the simulations used in the RF models' training, testing, and validation.

#### 3.3.2   Algorithm structure and setup

The algorithm has the following structure:

1. Calculation of the features with MiniRocket.

2. Selection of the AFlap fault symmetry (1B or 3B) and WT operative condition (NPP or PS).





3. Selection of the AFlap health states to be used in the classification.

4. Selection of the classifier: RF or Ridge.

5. For every specified wind speed range:

    (a) Training of the classification model:

        i. Training of one (or more) classification model.

        ii. If more than one classification model is trained, select the model with higher F1-score.

    (b) Validation of the trained classification model with the Val sets.

In the AFS method, we used the following sklearn python codes (Pedregosa et al., 2011): **StratifiedShuffleSplit** to create multiple Test and Training subsets from the TaT sets; **RandomForestClassifier** to train, test and validate the RF models; **RidgeClassifierCV** to train, test and validate the Ridge classifier models; **f1_score** to compute the F1-score; **StandardScaler** to standardize features by removing the mean and scaling to unit variance.

Starting from the setup proposed by (Abdallah, 2019), we investigate the optimal RF setup for the different scenarios, obtaining as optimal values: test rate of 30%, number of trees of 100, Shannon entropy as the criterion to measure the quality of a split, maximum depth of the tree of 5, minimum number of samples required to split an internal node of 5, and all features included. For the Ridge setup, the regularization strength parameter was set to an evenly spaced vector in log space of 1000 values between 1 and $10^6$, and the cross-validation set to Leave-One-Out Cross-Validation that handles efficiently the case of the number of features higher than the number of samples.

## 4 Manual feature selection results

The potential of the MFS approach in detecting a flap system's fault is investigated for several AFlap fault scenarios. These scenarios cover different combinations of AFlap fault symmetry (1B or 3B), WT operative conditions (NPP or PS), the possible split in different wind speed ranges, and different initial features selection (All or Reduced). The initial features option allows the reduction of the features used in training. We set the features subset without the Catch22 collection as the default Reduced setup. Instead, the wind speed ranges option enables the training of a dedicated ML model for every specified wind speed range. The default ranges used in this paper are Below Rated (BR: wind speed between 3.5 and 9.5 $ms^{-1}$), Around Rated (Rt: wind speed between 9.5 and 16.5 $ms^{-1}$), and Above Rated (AR: wind speed between 16.5 to 25 $ms^{-1}$).

Furthermore, we investigate three different fault detection levels:

– **Primary**: the model is trained to detect only the four primary health states: flap not active (AF_Off), active (AF_On), not active with fault (AF_Off_Fault), and active with fault (AF_Off_Fault).

– **Degraded**: the model is trained to detect if the flap has degraded performance but without identifying the performance degradation level. The three health states of flap with degraded performance are merged in a single state, called active with degradation (AF_On_Degr), that is included in the training with the four primary health states.



– **Detailed**: the model is trained to identify the flap performance reduced to 25% (AF_On_25pc), 50% (AF_On_50pc), and 75% (AF_On_75pc) in addition to the primary health states.

Table 2 collects the list of the MFS scenarios and shows their setup. Scenarios stated within parenthesis have a customized setup detailed described in the following chapters.

365 The selection of the models' performance metrics is strictly related to the requirements of the detection system. If one (or more) flap fault is critical for the WT integrity, the detection of this fault would be prioritized over the other AFlap health states. In this case, a good metric would be the Recall of the critical fault. For an opposite scenario, where it would be more critical to avoid false fault detections and keep the WT operating, the Precision of the different faults should be considered. In this paper, we are not considering any particular requirement for the fault detection system, and we aim to correctly detect all the 370 different classes equally without prioritizing anyone specifically. Therefore we select the F1-score, a trade-off between Recall and Precision that rewards the reduction of both false positives and false negatives. In detail, we use the weighted F1-score: the average of the F1-score of each class weighted by the ratio of the number of samples of each class over the total sample number. This metric is consistent between balanced and unbalanced classification tasks, allowing us to properly evaluate the few scenarios where the classes are not balanced.

375 **4.1 Detection of asymmetric fault**

Table 3 collects the performance of the RF models trained for the asymmetric fault scenarios described in Table 2. In addition to the weighted F1-score, the number of features obtained after the automatic feature reduction in the model training is shown. We use this number as an estimate of the model complexity: where more features are needed, more complex is to implement and execute the model. In addition, Precision and Recall values of the AFlap health states for some specific asymmetric fault 380 scenario models are collected in Table 4.

As the first step, scenario 1BN_A1 trains a single RF model for all the wind speeds to detect the Primary flap health states in normal power production, starting the training with all the available features. The trained model needs only 3 features (out of 400) to achieve an F1-score of 1, meaning it can perfectly classify the Primary health states. To understand how this trained model would perform with AFlap degradation (that will occur during the normal lifetime of the flap, also as a partial fault), 385 we test it with all the degraded AFlap health states classified as AF_On_Fault (scenario 1BN_A1c). Under this condition, the model F1-score decreases to 0.79 due to the misclassification of the flap fault (AF_On_Fault) as normal flap operation (AF_On). This misclassification is expressed numerically by a low value of AF_On Recall (0.33) and AF_On_Fault Precision (0.48). We obtain similar results in scenario 1BN_A2, where the wind speeds are split into three different wind speed ranges, and an independent RF model is trained for each range. The models trained in this scenario detect well the Primary health 390 states (F1-score of 1) but cannot distinguish the degraded AFlap states (scenario 1BN_A2c).

As the second step, we unify the degraded AFlap health states under a single category (AF_On_Degr). Scenario 1BN_B1 trains a single RF model for all the wind speeds to detect the Degraded fault health states in NPP. The trained model requires 17 features for an F1-score of 0.90, mainly due to the low Recall (0.56) of the AF_On class. Removing the Catch22 features





**Table 2.** Compact description of the setup of the AFlap fault scenarios.

| Fault scenario names for detection level: | | | Fault scenario setup: | | | |
|---|---|---|---|---|---|---|
| Primary | Degraded | Detailed | Features | Wind speed ranges | WT operation | Fault symmetry |
| 1BN_A1 (1BN_A1c) | 1BN_B1 | 1BN_C1 | All | No | NPP | Asymmetric (1B) |
| - | 1BN_B1r | - | Reduced | | | |
| 1BN_A2 (1BN_A2c) | 1BN_B2 | 1BN_C2 | All | Yes | | |
| - | 1BN_B2r | 1BN_C2r | Reduced | | | |
| - | 1BP_B1 | 1BP_C1 | All | No | PS | |
| 1BP_A1r (1BP_A1rc) | 1BP_B1r | 1BP_C1r | Reduced | | | |
| 3BN_A01 | - | - | All | No | NPP | Symmeatric (3B) |
| 3BN_A02 (3BN_A02c) | 3BN_B2 (3BN_B2b) | 3BN_C2 (3BN_C2b) | All | Yes | | |
| - | 3BN_B3 | 3BN_C3 | All | Yes | | |
| 3BP_A1 (3BN_A01c) | 3BP_B1 (3BP_B1b) | 3BP_C1 (3BP_C1b) | All | No | PS | |

to simplify the model (scenario 1BN_B1r) leads to a lower F1-score (0.82), poor AF_Off_Fault class Recall (0.41), and poor AF_On class Precision (0.37). Splitting the training for the tree wind speed ranges (scenario 1BN_B2) generates three high-performing models (F1-score higher than 0.95) even in the scenario where the Catch22 features are removed (1BN_B2r).

As the last step for the NPP case, we test if a model could individually identify the degraded AFlap health states. Scenario 1BN_c1 trains a single RF model for all the wind speeds for a Detailed detection level in NPP. The trained model requires 16 features for an F1-score of 0.70 and can almost not distinguish the AF_Off_Fault from the other classes. Splitting the training for the tree wind speed ranges (scenario 1BN_c2) dramatically improves the performance of the models with an F1-score of 0.91 BR, 0.95 around rated, and 0.98 AR obtained with 14 features or less. In detail, the BR model is imprecise in classifying the AF_Off_Fault and has a low Recall for AF_On. Removing the Catch22 features leads to models with similar performance but fewer features (10 or less).

After the NPP scenarios, we investigated if the AFlap health states can be correctly classified in pre-startup conditions where the WT is idling due to low wind speed. For the scenarios aiming at the Primary flap health states (1BP_A1r and 1BP_A1rc) in the PS condition, the performance follows the same pattern as the previous similar scenarios in NPP. When the AF_On_Degr class is included (scenario 1BP_B1), the trained model shows a high F1-score (0.94) with 20 features. A high F1-score of 0.95



**Table 3.** F1-score of the asymmetric flap fault scenarios evaluated with MFS and AFS approaches. The number of features used for the MFS is also specified.

| Blade Fault | WT operation | Wind speed range [$ms^{-1}$] | Basic | | | | Degraded | | | | Detailed | | | |
|---|---|---|---|---|---|---|---|---|---|---|---|---|---|---|
| | | | | MFS | | | | MFS | | AFS | | MFS | | AFS |
| | | | Case name | RF F1-Score | - | # Features | Case name | RF F1-Score | - # Features | RF F1-score | Case name | RF F1-Score | - # Features | RF F1-score |
| 1B | NPP | 3.5 - 25.5 | 1BN_A1 (1BN_A1c) | 1 (0.79) | - | 3 | 1BN_B1 | 0.90 | - 17 | 0.67 | 1BN_C1 | 0.70 | - 16 | 0.55 |
| | | | | | | | 1BN_B1r | 0.82 | - 11 | | | | | |
| | NPP | 3.5 - 9.5 (BR) | 1BN_A2 (1BN_A2c) | 1 (0.80) | - | 10 | 1BN_B2 | 0.96 | - 10 | 0.59 | 1BN_C2 | 0.91 | - 14 | 0.52 |
| | | 9.5 - 16.5 (RT) | | 1 (0.82) | - | 5 | | 0.98 | - 9 | 0.78 | | 0.95 | - 14 | 0.73 |
| | | 16.5 - 25.5 (AR) | | 1 (0.80) | - | 4 | | 0.99 | - 5 | 0.62 | | 0.98 | - 4 | 0.56 |
| | | 3.5 - 9.5 (BR) | | | | | 1BN_B2r | 0.95 | - 6 | | 1BN_C2r | 0.90 | - 10 | |
| | | 9.5 - 16.5 (RT) | | | | | | 0.90 | - 7 | | | 0.94 | - 10 | |
| | | 16.5 - 25.5 (AR) | | | | | | 0.98 | - 3 | | | 0.99 | - 6 | |
| | PS | 0 - 3.5 | | | | | 1BP_B1 | 0.94 | - 20 | 0.45 | 1BP_C1 | 0.92 | - 27 | 0.38 |
| | | | 1BP_A1r (1BP_A1rc) | 1 (0.80) | - | 5 | 1BP_B1r | 0.95 | - 9 | | 1BP_C1r | 0.93 | - 14 | |

is also achieved with only 9 features starting from a reduced set of features (scenario 1BP_B1r). Finally, when we analyze the Detailed detection level (scenario 1BP_C1), the trained model achieves an F1-score of 0.92 with 27 features. Omitting the
Catch22 features in training (scenario 1BP_C1r) brings an equivalent F1-score with only 14 features.

### 4.2 Detection of symmetric fault

Table 5 collects the performance of the RF models trained for the symmetric fault scenarios described in Table 2. Precision and Recall values of the AFlap health states for some specific symmetric fault scenario models are collected in Table 6.

Similarly to the asymmetric faults cases, we start with a scenario (3BN_A1) that trains a single RF model for all the wind
speeds to detect the Primary health cases in NPP, using all the available features initially. The trained model achieves an F1-score of 0.75 with 32 features. When tested with the degraded flap health states bundled together as AF_On_Fault (scenario 3BN_A1c), the F1-score decreased to 0.71 due to the misclassification of the fault (AF_On_Fault) as normal flap operation (AF_On). This misclassification is expressed numerically by a low value of AF_On Recall (0.33) and low Precision (0.61) of both AF_Off_Fault and AF_On_Fault. We obtained similar results in scenario 3BN_A2, where we trained an independent RF
model for each of the three different wind speed ranges. The models trained in this scenario detect better than the previous scenario the Primary flap health states, especially for AR (F1-score of 0.92) but cannot correctly distinguish the degraded AFlap states (scenario 1BN_A2c).

As the second step, we include the degraded AFlap health states under a single category (AF_On_Degr). Scenario 3BN_B2 trains an independent RF model to detect the Degraded flap cases in NPP for each of the three different wind speed ranges. The





**Table 4.** Precision and Recall values of each AFlap health state of a selection of scenarios of asymmetric flap fault evaluated with MFS and AFS approaches.

| Approach | MFS | | | | | | | | AFS | | | |
|---|---|---|---|---|---|---|---|---|---|---|---|---|
| WT operation | NPP | | | | | | PS | | NPP | | | PS |
| Scenarios | 1BN_A1c | 1BN_B1 | 1BN_B1r | 1BN_C1 | 1BN_C2 | | 1BP_B1r | 1BP_C1r | 1BN_B1 | 1BN_C1 | 1BN_B2 | 1BP_B1 |
| Wind speed range | All | All | All | All | BR | AR | All | All | All | All | RT | All |
| **F1 score** | 0.79 | 0.90 | 0.82 | 0.70 | 0.91 | 0.98 | 0.95 | 0.93 | 0.67 | 0.55 | 0.78 | 0.45 |
| Precision — AF_Off | 1.00 | 1.00 | 0.98 | 0.97 | 1.00 | 1.00 | 0.95 | 0.98 | 0.81 | 0.80 | 0.99 | 0.66 |
| Precision — AF_Off_Fault | 1.00 | 0.77 | 0.70 | 0.00 | 0.53 | 0.95 | 0.84 | 0.78 | 0.89 | 0.93 | 1.00 | 0.63 |
| Precision — AF_On | 1.00 | 0.95 | 0.37 | 0.68 | 0.92 | 0.97 | 0.90 | 0.93 | 0.50 | 0.59 | 0.61 | 0.28 |
| Precision — AF_On_Fault | 0.48 | 1.00 | 1.00 | 0.98 | 1.00 | 0.98 | 0.95 | 0.98 | 0.47 | 0.49 | 0.55 | 0.27 |
| Precision — AF_On_Degr | - | 0.81 | 0.87 | - | - | - | 0.98 | 0.96 | 0.73 | - | 0.90 | 0.62 |
| Precision — AF_On_25pc | - | - | - | 0.88 | 1.00 | 0.98 | - | - | - | 0.40 | - | - |
| Precision — AF_On_50pc | - | - | - | 0.66 | 0.88 | 0.99 | - | 0.90 | - | 0.30 | - | - |
| Precision — AF_On_75pc | - | - | - | 0.97 | 1.00 | 1.00 | - | 0.95 | - | 0.42 | - | - |
| Recall — AF_Off | 1.00 | 1.00 | 1.00 | 1.00 | 1.00 | 1.00 | 1.00 | 0.97 | 0.90 | 0.95 | 1.00 | 0.59 |
| Recall — AF_Off_Fault | 1.00 | 0.95 | 0.41 | 0.00 | 1.00 | 1.00 | 1.00 | 1.00 | 0.79 | 0.76 | 0.99 | 0.70 |
| Recall — AF_On | 0.33 | 0.56 | 0.56 | 0.44 | 0.65 | 0.99 | 0.88 | 0.83 | 0.71 | 0.58 | 0.85 | 0.41 |
| Recall — AF_On_Fault | 1.00 | 0.97 | 0.96 | 0.92 | 1.00 | 0.98 | 0.99 | 0.95 | 0.60 | 0.54 | 0.93 | 0.50 |
| Recall — AF_On_Degr | - | 0.98 | 0.99 | - | - | - | 0.90 | - | 0.56 | - | 0.57 | 0.35 |
| Recall — AF_On_25pc | - | - | - | 0.84 | 0.99 | 0.99 | - | 0.90 | - | 0.21 | - | - |
| Recall — AF_On_50pc | - | - | - | 0.47 | 0.84 | 0.94 | - | 0.87 | - | 0.48 | - | - |
| Recall — AF_On_75pc | - | - | - | 1.00 | 1.00 | 0.98 | - | 0.99 | - | 0.32 | - | - |

trained model shows a low F1-score of 0.41 for BR, 0.51 at rated, and 0.64 for AR. To improve the performance of the models, we explore the RF model hyperparameters setup. One of the most successful results is scenario 3BN_B2b, where we increase the number of trees from 100 to 300 and the maximum number of slit from 12 to 30. These changes lead to an increase of the F1-score of around 0.08, with the highest value achieved in the AR wind sped range with an F1-score of 0.72. This model still shows low Recall for AF_On and AF_On_Fault flap states and a low Precision for AF_On_Degr. As a final try to increase

the models' performance, we reduce the width of the wind speed ranges, obtaining: BRa (3.5 to 6.5 $ms^{-1}$), BRb (6.5 to 9.5 $ms^{-1}$), RTa (9.5 to 12.5 $ms^{-1}$), RTb (12.5 to 15.5 $ms^{-1}$), ARa (15.5 to 20.5 $ms^{-1}$), and ARb (20.5 to 25.5 $ms^{-1}$). This scenario (3BN_B3) leads to RF models with higher performance but only ARa and ARb models have the F1-scores higher than 0.7 (0.75 and 0.83, respectively). Both RF models have good Precision except for the AF_On_Degr class but a low Recall for AF_On, AF_On_Fault and AF_On_Degr.

Finally, we investigate if an RF model could individually identify the Detailed flap health states, obtaining models with poor





**Table 5.** F1-score of the symmetric flap fault scenarios evaluated with MFS and AFS approaches. The number of features used for the MFS is also specified.

| Blade Fault | WT operation | Wind speed range [$ms^{-1}$] | Basic Case name | Basic MFS RF F1-Score | - | Basic MFS # Features | Degraded Case name | Degraded MFS RF F1-Score | - | Degraded MFS # Features | Degraded AFS RF F1-score | Detailed Case name | Detailed MFS RF F1-Score | - | Detailed MFS # Features | Detailed AFS RF F1-score |
|---|---|---|---|---|---|---|---|---|---|---|---|---|---|---|---|---|
| 3B | NPP | 3.5 - 25.5 | 3BN_A1 (3BN_A1c) | 0.75 (0.71) | - | 32 | | | | | | | | | | |
| | NPP | 3.5 - 9.5 (BR) | 3BN_A2 (3BN_A2c) | 0.74 (0.69) | - | 27 | 3BN_B2 (3BN_B2b) | 0.41 (0.49) | - | 20 (23) | 0.53 | 3BN_C2 (3BN_C2b) | 0.24 (0.35) | - | 21 (22) | 0.49 |
| | | 9.5 - 16.5 (RT) | | 0.85 (0.75) | - | 31 | | 0.51 (0.63) | - | 18 (25) | 0.61 | | 0.30 (0.40) | - | 15 (16) | 0.55 |
| | | 16.5 - 25.5 (AR) | | 0.92 (0.81) | - | 15 | | 0.64 (0.72) | - | 8 (9) | 0.63 | | 0.50 (0.61) | - | 7 (7) | 0.58 |
| | NPP | 3.5 - 6.5 (BRa) | | | | | 3BN_B3 | 0.61 | - | 7 | 0.49 | 3BN_C3 | 0.48 | - | 7 | 0.43 |
| | | 6.5 - 9.5 (BRb) | | | | | | 0.55 | - | 16 | 0.55 | | 0.39 | - | 17 | 0.49 |
| | | 9.5 - 12.5 (RTa) | | | | | | 0.67 | - | 20 | 0.55 | | 0.48 | - | 15 | 0.49 |
| | | 12.5 - 15.5 (RTb) | | | | | | 0.62 | - | 9 | 0.64 | | 0.45 | - | 8 | 0.59 |
| | | 15.5 - 20.5 (ARa) | | | | | | 0.75 | - | 8 | 0.67 | | 0.58 | - | 9 | 0.60 |
| | | 20.5 - 25.5 (ARb) | | | | | | 0.83 | - | 7 | 0.49 | | 0.77 | - | 9 | 0.50 |
| | PS | 0 - 3.5 | 3BP_A1 (3BP_A1c) | 0.80 (0.71) | - | 12 | 3BP_B1 (3BP_B1b) | 0.55 (0.55) | - | 11 (22) | 0.50 | 3BP_C1 (3BP_C1b) | 0.30 (0.38) | - | 7 (7) | 0.43 |

performance in the symmetric fault case during NPP. Scenario 3BN_C2 trains an independent RF model to detect the Degraded fault cases in NPP for each of the three different wind speed ranges. The trained model has an F1-score below 0.50. Increasing the number of trees to 300 and the maximum number of slit to 30 (scenario 3BN_C2b) improves the F1-score of around 0.1. The Precision and Recall values show that the models cannot correctly detect most AFlap health states. Adding the reduction of the size of the wind ranges (scenario 3BN_C3) does slightly improve the F1-score, but with the best performing model (ARb) reaching only an F1-score of 0.77. For the PS wind turbine operation, the scenario aiming at the Degraded flap health states (scenario 3BP_B1) trains a model with a low F1-score (0.55) unable to classify all the different AFlap health states correctly. Increasing the number of trees and split (3BP_B1b) does not lead to better performance. Also, for the Detailed detection level (scenario 3BP_C1), the trained model achieves a poor F1-score of 0.30 that is only marginally improved (0.38) by increasing the number of trees and split (3BP_C1b).

## 5 Automatic feature selection results

The AFS approach is investigated with most of the scenarios used for the MFS approach and collected in Table 2. An initial preliminary investigation shows that a model trained only to detect the Primary flap health states will most likely perform poorly when the AFlap starts to have degraded performance, similar to what we obtain in the MFS analysis. Since this performance degradation is likely to happen, there is a low interest in a model that cannot account for it properly, and the Primary fault detection level is therefore omitted in the AFS analyses. Furthermore, the initial feature reduction does not apply to the AFS





**Table 6.** Precision and Recall values of each AFlap health state of a selection of scenarios of symmetric flap fault evaluated with MFS and AFS approaches.

| Approach | MFS | | | | | | | | | AFS | | | |
|---|---|---|---|---|---|---|---|---|---|---|---|---|---|
| WT operation | NPP | | | | | | | | PS | NPP | | | PS |
| Scenarios | 3BN_A1c | 3BN_B2b | | 3BN_B3 | | 3BN_C2b | | 3BP_C3 | 3BP_B1 | 3BN_B2 | 3BN_B3 | 3BP_C3 | 3BP_B1 |
| Wind speed range | All | BR | AR | ARa | ARb | BR | AR | ARb | All | AR | ARa | ARa | All |
| F1 score | 0.71 | 0.49 | 0.72 | 0.75 | 0.83 | 0.35 | 0.61 | 0.77 | 0.55 | 0.63 | 0.67 | 0.60 | 0.50 |
| **Precision** AF_Off | 0.84 | 0.44 | 0.68 | 0.80 | 0.94 | 0.37 | 0.81 | 0.78 | 0.06 | 0.94 | 0.90 | 0.93 | 0.74 |
| AF_Off_Fault | 0.61 | 0.58 | 0.77 | 0.74 | 0.94 | 0.39 | 0.44 | 0.78 | 0.15 | 0.95 | 0.98 | 0.98 | 0.78 |
| AF_On | 0.71 | 0.71 | 0.82 | 0.80 | 0.89 | 0.50 | 0.71 | 0.78 | 0.15 | 0.40 | 0.40 | 0.54 | 0.29 |
| AF_On_Fault | 0.62 | 0.73 | 0.94 | 0.84 | 1.00 | 0.15 | 0.41 | 0.78 | 0.10 | 0.42 | 0.53 | 0.53 | 0.35 |
| AF_On_Degr | - | 0.16 | 0.54 | 0.65 | 0.61 | - | - | - | 0.97 | 0.67 | 0.76 | - | 0.65 |
| AF_On_25pc | - | - | - | | | 0.09 | 0.49 | 0.78 | - | - | - | 0.38 | - |
| AF_On_50pc | - | - | - | | | 0.21 | 0.47 | 0.61 | - | - | - | 0.33 | - |
| AF_On_75pc | - | - | - | | | 0.63 | 0.78 | 0.83 | - | - | - | 0.29 | - |
| **Recall** AF_Off | 0.69 | 0.94 | 0.97 | 1.00 | 1.00 | 0.63 | 0.97 | 0.93 | 1.00 | 0.95 | 0.98 | 0.98 | 0.79 |
| AF_Off_Fault | 0.79 | 0.82 | 1.00 | 1.00 | 1.00 | 0.67 | 1.00 | 1.00 | 1.00 | 0.94 | 0.89 | 0.92 | 0.72 |
| AF_On | 0.32 | 0.30 | 0.51 | 0.52 | 0.64 | 0.37 | 0.53 | 0.74 | 0.60 | 0.79 | 0.85 | 0.78 | 0.55 |
| AF_On_Fault | 0.89 | 0.28 | 0.52 | 0.60 | 0.58 | 0.20 | 0.41 | 0.70 | 0.70 | 0.85 | 0.80 | 0.62 | 0.66 |
| AF_On_Degr | - | 0.58 | 0.78 | 0.77 | 0.91 | - | - | - | 0.45 | 0.22 | 0.35 | - | 0.26 |
| AF_On_25pc | - | - | - | | | 0.20 | 0.51 | 0.82 | - | - | - | 0.28 | - |
| AF_On_50pc | - | - | - | | | 0.20 | 0.40 | 0.58 | - | - | - | 0.31 | - |
| AF_On_75pc | - | - | - | | | 0.29 | 0.63 | 0.68 | - | - | - | 0.22 | - |

approach, and the scenarios with the Reduced feature setup are not included. Regarding the RF hyperparameters setup, we ran an initial study using several randomly generated subsets of features to identify the values of the hyperparameters optimizing the F1-score. This study shows an optimal hyperparameters setup with the number of trees of 100 (an increase up to 300 does

not improve the performance), the maximum depth of the tree of 5 (lower values tend to cause overfitting), the minimum number of samples required to split an internal node of 5, and Shannon entropy as a criterion to measure the quality of a split. The final model training is instead performed including all the features. This configuration showed better performance compared to the random pick of a subset of features of the size of square root or $log2$ of the total number of features.

Having a number of features considerably higher than the number of samples, a condition not ideal for the RF method, we

investigate if another classifier can perform better than RF. We selected the Ridge regression classifier with Cross Validation, a linear classifier tested in the development of both ROCKET (Dempster et al., 2020) and MiniRocket (Dempster et al., 2021).





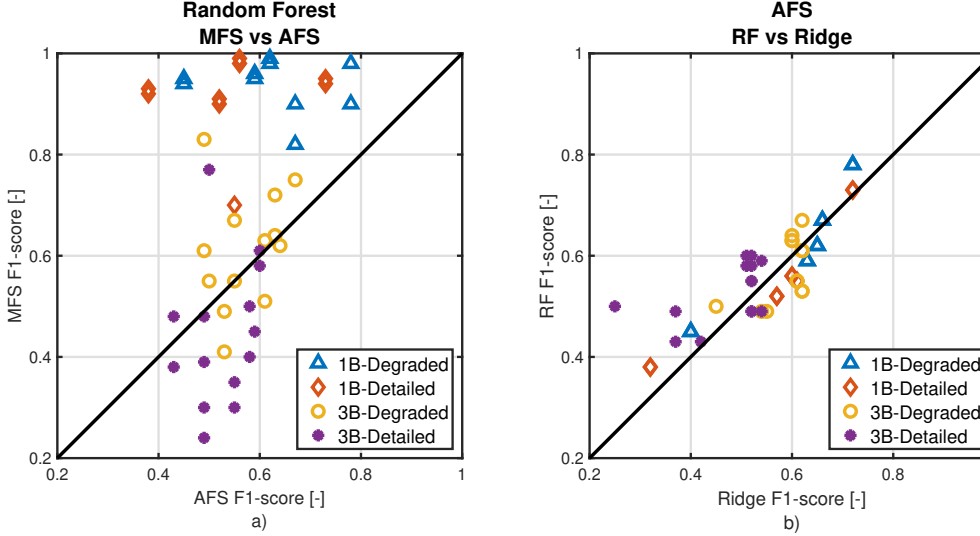

**Figure 6.** a) Comparison between the MFS RF F1-scores and the MFS RF F1-scores of the different Degraded and Detailed flap health scenarios. b) Comparison between the AFS RF F1-scores and the AFS Ridge F1-scores of the different Degraded and Detailed flap health scenarios.

## 5.1 Detection of asymmetric fault

The performances of models trained with the RF classifier for the asymmetric fault scenarios described in Table 2 are shown in Table 3, together with the results from the MFS approach. Table 4 collects Precision and Recall values of the AFlap health

states for some specific asymmetric fault scenarios of the RF models.

In NPP, AFS RF shows low performance when trained to detect the Degraded health states for all the wind speeds (scenario 1BN_B1) with an F1-score of 0.67. The model can correctly classify the AF_Off and AF_Off_Fault states but cannot classify the other AFlap health states, as indicated by the Recall and Precision values. When we train the classifiers for the Detailed health states (scenario 1BN_C1), performance decreases with an F1-score of 0.55. Also, for this scenario, the models can

adequately classify AF_Off and AF_Off_Fault states but fail with the other states. Performances slightly increase when we split the training into different wind speed ranges. AFS RF achieves the highest performance in evaluating the Degraded health states (scenario 1BN_B2) around rated wind speed with an F1-score of 0.78. For the other wind speed ranges, the F1-score stays around 0.6. Similar results are achieved for the Detailed health states (scenario 1BN_C2), where AFS RF achieves a max F1-score of 0.73 at RT and not higher than 0.56 in the other wind speed ranges. Looking at the results in PS, the RF classifier

performs poorly in identifying Degraded and Detailed health states (scenario 1BP_B1 and 1BP_C1, respectively) with a max F1-score of 0.45.

Figure 6 compares the F1-scores from AFS RF with the scores from AFS Ridge. The Ridge classifier achieves similar results




compared to RF for Degraded and Detailed flap health states in the 1B case. Also, the Precision and Recall values of the Ridge models are close to the values of the RF models of the corresponding scenarios. For brevity, we have not included them in this paper.

## 5.2 Detection of symmetric fault

The performances of models trained with the RF classifier for the symmetric fault scenarios described in Table 2 are shown in Table 5, together with the results from the MFS approach. Precision and Recall values of the AFlap health states for some specific asymmetric fault scenario models are collected in Table 6.

In NPP, AFS RF shows low performance when trained to detect the Degraded health states (scenario 3BN_B2) for different wind speed ranges. The F1-score is between 0.53 at BR and 0.63 at AR. Similarly to what was observed in the asymmetric fault, the models can detect most of the AF_Off and AF_Off_Fault states properly but cannot classify the other AFlap health states. Reducing the width of the wind speed ranges (scenario 3BN_B3) mainly reduces the performance, especially for low and high wind speeds, except at ARa, where the F1-score increases up to 0.67. Looking at the Detailed health states for the three wind seed ranges (scenario 3BN_C2), the F1-score rises from 0.49 at BR to 0.58 at AR. Similarly to the Degraded flap health states, reducing the width of the wind speed ranges (scenario 3BN_C3) does reduce the performance, especially for low or high wind speeds. Like the previous scenarios, the models have high Recall and Precision values for AF_Off and AF_Off_Fault states and low values for the other health states.

Looking at the results in PS, AFS RF classifiers perform poorly in identifying Degraded and Detailed health states (scenario 1BP_B1 and 1BP_C1 respectively) with a max F1-score of 0.5. Also, the ability to correctly classify AF_Off and AF_Off_Fault states is consistently reduced with both Recall and Precision below 0.8.

Similarly to the asymmetrical case, the Ridge classifier achieves similar results to RF for both Degraded and Detailed flap health states in the 3B case. Figure 6 shows that AFS RF performs slightly better than the Ridge classifier for the Detailed flap health states.

# 6 Discussion

## 6.1 Manual feature selection with random forest

The results described in Section 4.1 show that the manual feature selection approach with a random forest classifier can correctly classify the AFlap health states in the case of asymmetric flap fault 1B. In normal power production, Degraded and Detailed health states are correctly classified, and the performance increases when splitting the training into three wind speed ranges. This result supports our initial hypothesis that as a WT operates differently at different wind speed ranges, models trained for specific wind ranges perform better than a single model trained for all wind speeds. Notably, less than 20 features are needed for the models, a small fraction of the around 400 provided at the beginning of the training. This number can be further reduced to 10 or less by removing the Catch22 features without significantly reducing the models' performance. Even





if few features are specific to some scenarios, all scenarios share the blade-to-blade differences of the mean blade root bending moment (mainly of flapwise bending moment), followed by the mean value of WT performance indicators like pitch angle, generator power, rotor speed, or blade root bending moments. This sounds logical since an asymmetrical flap fault among the different blades should result in a relevant difference in the blades' loading. The blade-to-blade load difference channels should collect this load imbalance. Furthermore, the blade imbalance is a function of the WT operational working state that the models most probably identify with the WT performance indicator features. Also, the models need fewer features at above rated wind speed, where generator power and rotor speed are almost constant, and the blade-to-blade load difference is most likely less impacted by them. Looking at pre-startup operation, both Degraded and Detailed health states are correctly classified and the Catch22 features can be omitted without reducing performance, as experienced in NPP. The models need more features than the NPP scenarios, with still blade-to-blade load difference as main features followed by mean values of the blade loads. Generator power and rotor speed are no longer relevant, being almost null in the idling state.

Regarding the Primary health states detection, the models obtained with MFS show, on one side, high performance in identifying the four Primary health states in both NPP and PS, but on the other side, they cannot account properly for the degradation of the AFlap performance. Since this degradation is likely to happen in the WT lifetime, we think these models can lead to some significant misclassification and we do not recommend them for field application.

In the case of a symmetric flap fault 3B, the MFS approach fails to correctly classify all the tested AFlap health states in both NPP and PS operation states. Increasing the model complexity or reducing the wind range width improves the performance negligibly, and the only case with an acceptable F1-score is the Degraded health states at high wind speeds (ARb). Looking at the selected features helps to understand the reasons for the misclassification. The blade-to-blade features are no longer present, replaced by several features related to single-blade loading, tower top accelerations, and rotor speed. This result confirms that the blade-to-blade loads no longer contain any flap fault information in the symmetric fault scenario, where the flap fails symmetrically in all three blades. Therefore, the RF models try to estimate the AFlap health states from the features of other channels, like blade loads. The failure to obtain good performance means the channels used in training do not have sufficient information to identify the AFlap health states, and more signals are needed to achieve it. A possible solution to properly detect and classify AFlap health states in the 3B condition is to transform it into an asymmetrical case. This transformation can be achieved with a flap check routine that activates the flap one blade at a time, which is like a 1B condition where the RF models can accurately estimate the flap health states.

## 6.2 Automatic feature selection with random forest and ridge classifier

The results described in Section 5 show that the automatic feature selection approach with a random forest classifier cannot correctly classify the AFlap health states for both asymmetric and symmetric flap fault cases. The trained models do not reach an F1-score higher than 0.8 in 1B scenarios and higher than 0.7 in 3B scenarios. Figure 6 compares the F1-scores between MFS and AFS RF models, with the MFS models outperforming the AFS models in the 1B scenarios. In the 3B scenarios, AFS RF models perform slightly better, especially for the Detailed flap health states. As shown in Figure 6, for the 1B scenarios, AFS Ridge models perform similarly to the AFS RF models in 1B cases and slightly worst in the 3B cases. The overall





performances of the AFS models need to be improved for the AFS models to be implemented in detecting all the AFlap health states. However, for the NPP operation state, the AFS models can correctly identify the AF_Off and AF_OFF_Fault flap health
states from the other states with Precision and Recall above 0.9. If these two states are relevant to the WT design, the AFS models can be implemented to detect these two specific flap health states. Furthermore, this capability shows that the AFS approach has some potential that can be further explored with other ML techniques like, for example, Multirocket (MiniRocket evolution) or HYDRA.

## 7 Conclusions

In this paper, we investigated two approaches to identify the health state of a WT's active trailing edge flaps. These approaches do not rely on specific sensors designated for AFlap's health monitoring but only on sensors commonly available on all commercial wind turbines. Both approaches are based on multivariate time-series classification methods. The first method (MFS) uses manual feature engineering in combination with a random forest classifier. The second method (AFS) creates the feature vectors from MTS data by passing the inputs through multiple random convolutional kernels in combination with a random
forest classifier. We trained both methods to classify combinations of seven AFlap health states for a WT operating in normal power production and pre-startup. We analyzed asymmetrical flap faults, with the flap health states applied to only one blade, and symmetrical flap faults, where the flap health states were applied to all three blades. The study is based on a pool of aeroelastic simulations of a WT equipped with an active flap. These simulations were performed with a broad set of environmental conditions to account for the variability due to external weather conditions in the model's training. To keep the approach as
general as possible, we focused on identifying the AFlap health state when the flap is in stationary actuation positions. This approach keeps the detection system independent from any specific AFlap controller strategy, AFlap system design, or fault dynamics. The underlying idea is to integrate the monitoring system in an AFlap status check routine running for several minutes where the performance of the stationary flap is verified.

In this paper, we showed that the MFS method could classify the different combinations of AFlap health states in the case
of asymmetrical flap faults. The MFS method is reliable when the WT operates in normal power production and pre-startup, achieving an F1-score higher than 0.9. Essential features to achieve this result are the blade-to-blade differences of the mean blade root loads.

Instead, the MFS method failed to classify the AFlap health states in the case of symmetrical flap fault. This failure is likely due to the channels used for the training not providing sufficient information about the flap fault. To avoid adding other sensor
signals to the model, we suggest transforming the symmetrical flap fault detection into an asymmetrical one. For example, a flap check routine can activate the flap one blade at a time, generating a temporary asymmetrical flap activation that the MFS methodology can monitor.

Furthermore, we showed that the AFS method fails to classify most AFlap health states in asymmetrical and symmetrical flap faults. However, for both cases, with the wind turbine in normal power production, the AFS method can be used to identify
two specific flap health states.





We also tested a Ridge classifier in the AFS method, obtaining a similar performance to the random forest classifier with a consistently lower training time.

As future developments, we suggest further exploring the AFS method by applying different and more performing convolutional techniques. It is also of extreme interest to validate the capability of the MFS method with data from an actual wind
turbine, to which the models can be adapted via transfer learning techniques.

*Author contributions.* AG and IA conceptualized and designed the study. AG designed the objectives, performed analysis, and wrote the original draft paper. AI supported the methodology, the analysis and reviewed and edited the whole paper.

*Competing interests.* AG is hired by Siemens Gamesa Renewable Energy, company that is developing the flap technology used as reference in the paper.

*Acknowledgements.* The authors thank Gregory Duthé and Eleni Chatzi for their support. This research was partially funded by Danmarks Innovationsfond, Case no. 9065-00243B, PhD Title: "Advanced model development and validation of Wind Turbine Active Flap system" and by Otto Mønsteds Fond, application file number 22-70-0210. The authors also thank xxx and xxx anonymous reviewers whose comments and suggestions helped improve and clarify this manuscript.



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
