# Peer review of "Active Trailing Edge Flap System fault detection via Machine Learning"

_Wind Energy Science, 2023_

## Author Response (AR1)

**We thank the reviewers for taking the time to read our work. We here provide our answers and rebuttals.**

**Reply to RC1**

Dear Anonymous Referee #1,

Thank you for your comments.

- **Lack of clarity regarding the problem addressed:** we agree the specific real-world problem and benefits are not clearly stated. We have now modified the text in the abstract (lines 1 to 5 and 12 to 14), introduction (lines 24 to 34) and conclusion (lines 573 to 575) to add more clarity to the subject matter. Regarding previous research on the topic, we could not find any relevant papers. Therefore, in the introduction (lines 34 to 58) we described three approaches commonly used for detecting and monitoring faults on wind turbine components.
- **Incomplete analysis of AFS method performance:**
  - The actuator control signal is already included in the input signal used for the generation of the features (line 176)
  - Several parameters can affect the performances of the AFS. Due to time and resource constrains, we just want to show that the AFS method has some potential and, in some configurations, performs better than MFS. But it needs to be further investigated and developed to be usable. We added this consideration in the AFS discussion (lines 562 to 568) and in the conclusions (lines 607 to 609).
- **Potential improvement:** Ensemble modelling is a technique that has been used by the authors of this paper and can confirm the reviewer's comment that it very well might be an ideal approach to fusing the strength of multiple methods simultaneously. We will note this improvement in our future research. Due to time and resources constrains, it cannot be evaluated in this current version of the paper. The idea has been added as possible future development (lines 616 and 617).

**Reply to RC2:**

Dear Davide Astolfi,

Thank you for taking the time to review our work. Here are our responses to your questions:

- **Time resolution:** The data are obtained from aeroelastic simulations of 10 minutes length with a time resolution of 0.01 s. (line 137).
- **objective accomplishable for on site wind turbine with standard SCADA:** We developed this methodology taking into consideration it should be applicable to actual commercial wind turbines. The Manual Feature Selection approach with

reduced set of features relies only on 10 minutes statistical properties of commonly available wind turbine signals. Therefore, we believe this approach can be directly implemented to an actual prototype. The model must be trained with simulations based on the target wind turbine aeroelastic model and eventually tuned with transfer learning techniques using the wind turbine SCADA data. The MLS method with full features requires the calculation of additional features generally not included in the standard SCADA data. A cost benefit evaluation should be performed to decide which features are relevant to be captured in addition to the SCADA data. The Automatic Features Selection methodology does not use SCADA data. It requires instead the special postprocessing of the 10 minutes high frequency sampled time series. We included this consideration in the discussion (lines 546 to 551 and 569 to 571) and conclusion (lines 599 to 604 and 611 to 612) chapters.

**Paper updates:**

[revised manuscript text omitted]

---

## Referee Report (RR1)

**Referee report**

**General quality**

**Scientific significance: Excellent**

- The goal of the paper (to see whether it is possible to predict problems with the AFlaps using readily available SCADA data and simple ML models) seems to me to be relevant. SCADA data is readily available, which means it can be used at no additional cost. This makes it especially relevant for the industry. Furthermore, the fact that this research focusses on simple ML models first is useful because it gives an idea of what the lower-bound is of the modelling complexity.

**Scientific quality: Good**

- The research quality is good. The methodology seems to be sound. I could not find any clear methodological flaws or shortcomings. The methodology is well described in the manuscript. However, the manuscript might benefit from a schematic overview of the methodology. The results are well presented. Both the positives and negatives of the results are pointed out by the authors, which is a good thing because it gives a better understanding of the performance of the methodology. The literature study is however limited. Only a small number of papers are mentioned even though condition monitoring of wind turbines is currently a hot research topic. More recent papers that give an overview of techniques that are used for condition monitoring of wind turbines exist.

**Presentation quality: Good**

- The figures are clear. The result tables can, due to the usage of abbreviations for the different cases or scenarios, be somewhat difficult to understand at a glance. This is to a certain extent solved by table 2. The text however does contain multiple typo's and sentence errors.

**Suggestions for revision**

1. The addition of a schematic overview of the methodology to the manuscript. This will make it more clear for the reader what the different steps of the methodology are.
2. Expansion of the literature study on page 2 with more recent (overview) papers on condition monitoring of wind turbines.
3. Check the text to make sure that all abbreviations are at their first appearance preceded by their meaning. See for example the abstract.
4. Check the text on typo's and sentence errors.
5. Line 250: … k the ridge coefficients to be minimized. Where in Equation 7 is k? Please clarify in the manuscript.
6. Give a more thorough/in-depth explanation why some methodological decisions were taken: feature generation techniques, feature selection techniques, used models. For example why was MiniRocket selected and not a different feature engineering method? Why was the random forest selected instead of other simple models like SVM, ...? Please clarify in the manuscript.
7. Typo in line 367: …, and active with fault (AF_Off_Fault) -> should it not be AF_On_Fault? Please clarify in the manuscript.
8. It might be useful to add a table with abbreviations to the paper so that it is easier to look up the meaning.

---

## Author Response (AR2)

Dear Anonymous referee #3,
Thank you for your review.

I have addressed your comments as following (all lines refer to the
FlapFaultDetectionModel_Article_rev3TrackChanges file) :

1. The addition of a schematic overview of the methodology to the manuscript. This will make it more clear for the reader what the different steps of the methodology are. AG: Figure of the 2 methodologies have been added in the new figure 6 and 7
2. Expansion of the literature study on page 2 with more recent (overview) papers on condition monitoring of wind turbines. AG: Added references of review papers in chapter 2
3. Check the text to make sure that all abbreviations are at their first appearance preceded by their meaning. See for example the abstract. AG: Abbreviations definitions corrected.
4. Check the text on typo's and sentence errors. AG: Text corrected by some typos and grammatical errors
5. Line 250: … k the ridge coefficients to be minimized. Where in Equation 7 is k? Please clarify in the manuscript. AG: k corrected with w
6. Give a more thorough/in-depth explanation why some methodological decisions were taken: feature generation techniques, feature selection techniques, used models. For example why was MiniRocket selected and not a different feature engineering method? Why was the random forest selected instead of other simple models like SVM, ...? Please clarify in the manuscript.
AG:
-) Improved explanations about the 2 methodologies approaches in lines 117 to 119, 213 to 214
-) Improved explanation on the choice of MiniRocket algorithm from line 266 to line 275
-) Reasons for choosing random forest are stated in lines 254 to 264
7. Typo in line 367: …, and active with fault (AF_Off_Fault) -> should it not be AF_On_Fault? Please clarify in the manuscript. AG: typo corrected
8. It might be useful to add a table with abbreviations to the paper so that it is easier to look up the meaning. AG: added a list of Symbols from line 15 to 40

Best Regards,
Andrea Gamberini